# Step-by-Step Immune Activation for Suicide Gene Therapy Reinforcement

**DOI:** 10.3390/ijms22179376

**Published:** 2021-08-29

**Authors:** Irina Alekseenko, Alexey Kuzmich, Liya Kondratyeva, Sofia Kondratieva, Victor Pleshkan, Eugene Sverdlov

**Affiliations:** 1Institute of Molecular Genetics of National Research Centre “Kurchatov Institute”, 123182 Moscow, Russia; akrubik@gmail.com (A.K.); vpleshkan@gmail.com (V.P.); 2Shemyakin-Ovchinnikov Institute of Bioorganic Chemistry of the Russian Academy of Sciences, 117997 Moscow, Russia; liakondratyeva@yandex.ru (L.K.); sofia.a.kondr@gmail.com (S.K.); 3Institute of Oncogynecology and Mammology, National Medical Research Center for Obstetrics, Gynecology and Perinatology Named after Academician V.I. Kulakov of the Ministry of Healthcare of Russian Federation, 117997 Moscow, Russia

**Keywords:** cancer, tumor, immunosuppression, suicide gene therapy, immunotherapy, GM–CSF

## Abstract

Gene-directed enzyme prodrug gene therapy (GDEPT) theoretically represents a useful method to carry out chemotherapy for cancer with minimal side effects through the formation of a chemotherapeutic agent inside cancer cells. However, despite great efforts, promising preliminary results, and a long period of time (over 25 years) since the first mention of this method, GDEPT has not yet reached the clinic. There is a growing consensus that optimal cancer therapies should generate robust tumor-specific immune responses. The advent of checkpoint immunotherapy has yielded new highly promising avenues of study in cancer therapy. For such therapy, it seems reasonable to use combinations of different immunomodulators alongside traditional methods, such as chemotherapy and radiotherapy, as well as GDEPT. In this review, we focused on non-viral gene immunotherapy systems combining the intratumoral production of toxins diffused by GDEPT and immunomodulatory molecules. Special attention was paid to the applications and mechanisms of action of the granulocyte-macrophage colony-stimulating factor (GM–CSF), a cytokine that is widely used but shows contradictory effects. Another method to enhance the formation of stable immune responses in a tumor, the use of danger signals, is also discussed. The process of dying from GDEPT cancer cells initiates danger signaling by releasing damage-associated molecular patterns (DAMPs) that exert immature dendritic cells by increasing antigen uptake, maturation, and antigen presentation to cytotoxic T-lymphocytes. We hypothesized that the combined action of this danger signal and GM–CSF issued from the same dying cancer cell within a limited space would focus on a limited pool of immature dendritic cells, thus acting synergistically and enhancing their maturation and cytotoxic T-lymphocyte attraction potential. We also discuss the problem of enhancing the cancer specificity of the combined GDEPT–GM–CSF–danger signal system by means of artificial cancer specific promoters or a modified delivery system.

## 1. Introduction

The complexity and heterogeneity of malignant tumors make them highly robust to therapeutic treatment due to the emergence of drug resistance, the ability of the tumor to avoid immune surveillance, and the inability to increase concentrations of therapeutics due to their side effects. Therefore, most malignant diseases are treated with a combination of surgery, radiation therapy, and chemotherapy, which are necessary to destroy cancer cells and metastases. Cytotoxic chemotherapy kills rapidly dividing cells—mainly tumor cells but also normal ones. Such chemotherapy is most effective when used with combinations of different agents, often leading to synergistic effects. Combinations of drugs acting via different molecular mechanisms and on different signal pathways would be preferable for this goal. These combinations would increase the effectiveness of killing tumor cells and reduce the likelihood of resistance to drugs.

The emerging understanding that the targets of therapeutic effects can encompass not only the cancer cells themselves but also their interactions with the tumor microenvironment (TME) has led to revolutionary shifts in strategies for anticancer therapy. The successful use of T-cell checkpoint inhibitors in clinic has led to conceptual developments in the field of cancer immunology over the past two decades [1,2,3,4,5]. However, despite the evident success of this therapy in several cancer-afflicted individuals, a large number of patients with advanced malignancies do not benefit from such a treatment [6,7,8]. In addition, such treatment is accompanied by severe side effects caused by the development of numerous autoimmune inflammatory conditions [9]. Therefore, many clinical trials are underway to test the synergistic effects of combining immunotherapy with other therapies.

In this review, we highlight the methods for enhancing gene immunotherapy systems that combine the intratumoral production of both diffusing toxins and immunomodulatory cytokines, such as the granulocyte-macrophage colony stimulating factor (GM–CSF), with checkpoint immunotherapy and other anticancer immune response activators.

Such systems are expected to do the following:Act on different types of cancer, similar to classical chemotherapy or radiotherapy;Offer enhanced efficiency and safety due to their expression within the tumor, thus causing minimal damage outside the tumor and, at the same time, inducing overall antitumor and antimetastatic immunity;Remain inexpensive due to their simple production technology;Be used in combination with traditional chemo- and radiotherapy, as well as with newly developed immunotherapies.

## 2. GDEPT: Why Good Intentions Do Not Lead to Paradise in Suicide Cancer Gene Therapy

Gene-directed enzyme prodrug gene therapy (GDEPT) is a type of suicide gene therapy targeted at the systems common to all cancer cells—usually the replication system (Figure 1) [10,11,12,13,14,15,16]. In this regard, the GDEPT approach resembles chemotherapy but hits its targets from within cancer cells and is usually less toxic to normal cells. GDEPT is based on the delivery, into cancer cells, of genes encoding enzymes that can metabolize a separately administered low toxic prodrug into a cytotoxin, which kills the host cell but also diffuses into neighboring cells and kills them. This so-called bystander effect may greatly increase the efficiency of GDEPT.

Numerous enzyme–prodrug systems have been considered in various reviews (see [17,18,19]). Several of these systems are now in preclinical and clinical trials, including the most extensively studied systems of the herpes simplex virus thymidine kinase gene (HSVtk), with ganciclovir (GCV) as a prodrug, and the cytosine deaminase gene (CD) of Escherichia coli and yeast, which converts the prodrug antifungal agent 5-fluorocytosine (5-FC)—which features low toxicity—into the antineoplastic antimetabolite toxic 5-fluorouracil (5-FU) (See Appendix A). The mechanism of cell death caused by HSVtk plus GCV has been widely attributed to apoptosis [14]. Recently, it has become clear that HSVtk-GCV-mediated tumor cell lysis elicits antitumor immunity [20,21,22], which can eliminate tumor cells that do not express the suicide gene. This phenomenon could lead to the destruction of metastases originating from a primary tumor [12]. Foreign antigens, such as the HSVtk released after cell death, could play the same roles as viral antigens such as PAMP, as discussed below. An immune reaction directed towards cells expressing CD, independent of 5-FC, has also been reported [23]. This feature might be beneficial in terms of vaccination, but in the context of suicide gene therapy, it might lead to the early removal of CD-expressing cells [23]. Antitumor immunity activation was also observed in the cyclophosphamide GDEPT system [24,25].

The first gene therapy trial using GDEPT system HSVtk/GCV was approved in 1991 [26]. Since then, a lot of clinical studies have been undertaken (Appendix A). At least three drugs based on HSVtk/GCV and CD/5-FC systems have reached the third phase of clinical trials: Cerepro^®^ (developed by Ark Therapeutics Group, Kuopio, Finland), Toca-511 (developed by Tocagen, San Diego, CA, USA), and ProstAtak^®^ (developed by Candel Therapeutics, Inc., Needham, MA, USA).

Cerepro^®^ (sitimagene ceradenovec) is an adenoviral vector encoding the herpes simplex thymidine kinase gene, for the treatment of patients with high-grade glioma, and it received the status of an orphan drug in 2002. In Phase I/II of clinical studies, sitimagene ceradenovec exhibited a significant increase in survival. Although the preliminary results of a Phase III clinical study demonstrated a significant positive effect of sitimagene ceradenovec treatment on time to reintervention or death when compared with standard care treatment (hazard ratio: 1.43; 95% CI: 1.06–1.93; *p* < 0.05), the European Committee for Medicinal Products for Human Use did not consider the data to provide sufficient evidence of clinical benefit [27,28].

Toca 511 (vocimagene amiretrorepvec) is a nonlytic, retroviral replicating vector encoding the yeast cytosine deaminase. Toca 511 has shown highly promising results in early-stage clinical trials. In 2017, the European Medicines Agency granted the Toca 511 a priority review status, and the FDA designated it a “Breakthrough Therapy”. However, treatment with Toca 511 did not improve overall survival compared with the standard therapy in patients with recurrent high-grade glioma undergoing resection, missing the primary end-point of the phase III Toca 511 trial. Results showed that the median overall survival was 11.5 months and 12.2 months with Toca 511 and standard therapy, respectively (HR, 1.06; *p* = 0.6154). Moreover, all secondary endpoints did not show meaningful difference between the study arms [29]. Candel Therapeutics, Inc. is currently conducting a Phase 3 trial for ProstAtak^®^ (CAN-2409) in combination with the prodrug valacyclovir in patients with newly diagnosed localized prostate cancer who have an intermediate or high-risk for progression. It expects to complete enrollment in the 2021 with a final data readout in 2024. Thus, despite the great efforts, promising preliminary results, and long period time (over 25 years) since the conceptualization of this method in 1986 [30,31], as well as significant progress in preclinical studies and clinical trials, none of the various suicide gene therapy protocols have reached the clinical setting until nowadays [14,32]. Reasons for this lack of progress likely include a slow prodrug–drug conversion rate; the low transfection/transduction efficiency of the vectors; and nonspecific toxicity/immunogenicity related to delivery systems, plasmid DNA, enzymes, and/or prodrugs [10,15,33]. These barriers will require a combination of several strategies to overcome. In this review, we consider possible methods for increasing the effectiveness of GDEPT systems based on recent advances in understanding the developmental mechanisms of the immune response, which have recently led to great successes in cancer immunotherapy.

Previous researchers suggested enhancing the efficiency of GDEPT by using a combination of the suicide gene and the gene encoding an immune-activating cytokine. Many cytokines activate the immune system, including the granulocyte–macrophage colony-stimulating factor (GM–CSF), which was found to be one of the most potent inducers of anti-tumor activity in a variety of preclinical studies [34]. The combination of HSVtk in the adenoviral vector with another adenovirus containing both GM–CSF and IL-2 genes was also tested. The authors claimed that the co-expression of GM–CSF and IL-2 could augment the effects of HSVtk suicide gene therapy [35]. Other studies also confirmed efficiency of using two cytokines (IL-2 and GM–CSF) in combination with HSVtk in adenoviral vectors [36].

As indicated above, a combination of GM–CSF and HSVtk gene therapy showed greater therapeutic effects than HSVtk alone. The successful antitumor role of GM–CSF in these cases was explained by the enhancement of antigen uptake and the appearance of antigen-presenting cells after tumor cells are destroyed and release tumor-specific antigens [37,38]. In our study [10], we treated animals with an HSVtk/GM–CSF combination in one vector alongside a PEG-PEI-TAT peptide (PPT) copolymer for delivery. In model experiments, this treatment inhibited tumor growth and metastases, as well as increased lifespan. The data demonstrated that efficient antitumor effects could be achieved by using a simple and nonviral carrier with low toxicity to facilitate delivery of the vector containing both the suicide and immunomodulating genes.

However, GM–CSF is suspected to play a role in the appearance of the immune suppressive phenotype that promotes tumor progression and represents a significant impediment in successful therapy for metastatic cancer [39,40,41]. The problem of the balance between the immune-activating and immuno-inhibitory roles of GM–CSF remains far from resolved. Some of the relevant aspects of this issue are discussed in the next section.

## 3. A Brief Description of GM–CSF, the Most Commonly Used Immunomodulator

GM–CSF is most often used as an immunomodulator in various antitumor therapies and can serve as a paradigmatic example of attempts to achieve nonspecific activation of the immune environment of the tumor. GM–CSF is a cytokine belonging to a group of colony stimulation factors (CSFs) that are produced at a low level by most tissues and cell types but whose production can be increased up to 1000-fold by microorganisms, endo-toxins, or foreign cells [42]. GM–CSF can promote the growth, proliferation, and differentiation of numerous precursors and mature cell types. Knockout studies have demonstrated the largely redundant role of GM–CSF in maintaining granulocyte and macrophage populations in a physiologically steady state [43]. GM–CSF is currently considered a critical factor for dendritic cell (DC) development [44,45,46]. GM–CSF was the first cytokine shown to efficiently promote DC development in vitro and is used to induce DC differentiation from human monocytes and hematopoietic progenitor cells [45]. GM–CSF was also suggested to enhance DC activation and migration [47]. This enhancement may explain the successful antitumoral role of GM–CSF in combination with suicide genes: When tumor cells are destroyed and release tumor-specific antigens due to suicide-gene-mediated lysis, the presence of GM–CSF in the tumor microenvironment enhances antigen uptake and presentation [37]. This hypothesis is in line with the highly efficient antitumor action of oncolytic viruses armed with GM–CSF [48], which may be governed by the same effect, where the oncolytic virus destroys the cancer cells and delivers tumor antigens allowing GM–CSF to enhance antigen-presenting cells. In this way, a combination of cancer-cell-destroying agents with GM–CSF could be efficient in treating cancer and metastases.

At the same time, the immunomodulatory properties of GM–CSF play a significant role in the development of autoimmune diseases—for example, in maintaining immunological tolerance and regulating immune response. GM–CSF, moreover, has various pro-inflammatory functions and is involved in the development of autoimmune and inflammatory diseases. Further, the inhibition of GM–CSF in some animal models of autoimmune diseases showed significant beneficial effects. GM–CSF-producing helper T cells (Th17 cells) have also been identified to play a nonredundant role in the initiation of autoimmune inflammation, and the pathogenicity of Th17 cells has been associated with GM–CSF production [49]. The production of GM–CSF by T cells has also been associated with some autoimmune diseases, such as multiple sclerosis, rheumatoid arthritis, and myocarditis [50]. However, T, NK, and B cells do not express the GM–CSF receptor, thus excluding any direct effect of GM–CSF on lymphoid cells [51]. GM–CSF may modulate the function of multiple cell types and affect tolerance and autoimmunity in complex and presently unappreciated ways [43], likely through intermediate cell types or by acting in combination with other cytokines.

GM–CSF has been used for the treatment of malignancies as a monotherapy and as a vaccine adjuvant. These treatments have demonstrated contradictory outcomes that are sometimes beneficial but often lead to immune suppression and adversely affect the results of treatment [52].

There are numerous indications that GM–CSF is often upregulated in multiple types of human cancers and may be an important regulator of inflammation and immune suppression within the tumor microenvironment (TME) [39]. However, existing data are contradictory. Some studies suggest that GM–CSF inhibits tumor growth and metastasis (for example, in the case of colorectal cancer [53]), whereas other studies show that GM–CSF promotes tumor progression by supporting immunosuppressive TME and stimulating tumor growth and metastasis [54,55,56]. This diversity of data is characteristic of a complex system where minimal changes in the initial conditions can dramatically change the final result of the system’s development [57,58]. Thus, GM–CSF’s multifunctionality in combination with the complexity of cancer can lead to unpredictable effects, which may explain the contradictory outcomes when GM–CSF is used for cancer therapy. Additional immune system “enhancers” could improve therapeutic efficiency.

## 4. The Involvement of Danger Signals and Immunogenic Cell Death Could Further Improve the Antitumor Immune Response

One promising approach to facilitate the tumor inhibitory effect of GM–CSF is using “danger signals”. The “danger theory” of immune response proposed by Matzinger in 1994 (see [59,60]) maintains that “the immune system is more concerned with damage than with foreignness, and is called into action by alarm signals from injured tissues, rather than by the recognition of nonself”. This theory also argues that “APCs (antigen presenting cells) are activated by danger/alarm signals from injured cells, such as those exposed to pathogens, toxins, mechanical damage, and so forth” [60]. A number of relevant endogenous danger signals have been discovered [61]. Such endogenous molecules are normally invisible to the host immune system. However, when emitted by dying cells, these molecules initiate danger signaling and are known as “damage-associated molecular patterns” (DAMPs) [62,63]. DAMP chemistry varies greatly; they can be proteins; DNA, RNA, or metabolic products; ATP, uric acid, and heparin sulfate; or mitochondria [64]. DAMPs are detected by dedicated receptors known as “pattern-recognition receptors” (PRRs), such as toll-like receptors that cause intracellular and microenvironmental danger response [62] activating genes that code for inflammatory mediators coordinating the elimination of pathogens and damaged or infected cells. In addition to endogenous DAMP, these systems are also able to recognize the components of pathogenic organisms (bacteria and viruses) known as “pathogen-associated molecular patterns” (PAMPs).

In light of the above factors, one plausible way to enhance GM–CSF therapy effect may be to use DAMPs or PAMPs. Indeed, the enhanced production of GM–CSF in response to danger signals has been reported [65]—i.e., the stronger the danger is, the greater the chance that GM–CSF will work against tumor immune tolerance and in favor of destroying tumors because tumors are a source of danger. The introduction of genes encoding PAMPs in addition to GDEPT systems may represent another way to improve this response. An example of this approach was reported recently [66], although this study did not apply the GDEPT approach. The authors transfected tumor cells in vivo with genes of early secretory antigenic target-6 (ESAT-6) from *Mycobacterium tuberculosis*. This construct was injected intratumorally into syngeneic tumor-bearing mice and demonstrated significant tumor growth suppression.

Thus, the use of danger signals and GM–CSF will enhance the attracting antigen-presenting cells. Moreover, the formation of an antitumor response depends not only on these factors but also on the availability of tumor antigens. Tumor antigens are released during cancer cell death, which can be simultaneously immunogenic and non-immunogenic. Immunogenic cell death (ICD) causes changes in the cell surface composition, as well as the release of soluble substances. It was previously demonstrated (see [60]) that endogenous entities can initiate an immune response similar to DAMP signals. These DAMP-like signals operate on dendritic cell receptors and stimulate the presentation of tumor antigens to T cells. ICD leads to the activation of the immune system against cancer. Conversely, the inability to induce ICD, thus preventing the emission of immunogenic signals, or abolish the perception of ICD by the immune system can all contribute to therapeutic failure [62,67,68]. The immunogenic cell death can be ameliorated by some oncolytic viruses (OVs) [64,69,70] and HSVtk suicide gene therapy [71].

ICD releases tumor-associated antigens (TAAs). Many of these TAAs are absent from the cell surface, so T-cell receptors cannot detect them before cell death [72]. In addition, the lysis of cancer cells can release neoantigens that were not previously presented to the immune system and appeared in cancer cells due to, for example, somatic mutations [73,74]. Neoantigens may be able to activate dendritic cells and cause adaptive antitumor responses [75]. One interesting approach to increase danger signaling is to use “artificial neoantigen” in viral or bacterial danger signals or viral antigens in cancer vaccines. The corresponding antigens mimic the pathogenic particles that provide danger signals and facilitate uptake by dendritic cells. To systematize and help in the selection of danger signals, the National Cancer Institute compiled a list of potentially useful tumor antigens using several criteria, such as immunogenicity, therapeutic function, and tumor specificity [76,77].

Below, we consider the contributions of different danger signals to the efficiency of various tumor therapies:

The external induction of “danger signals” in cancer cells via the use of host-defense peptides. The stimulation of cancer cells with the antimicrobial peptide LTX-302 causes the release of several DAMP-signals, such as cytochrome-C, ATP, and HMGB1 [78], and the intratumoral injection of LTX-302 peptide induces an antitumor immune response in an experimental model of mouse melanoma [79].

The use of exogenous danger signals as TLR agonists. The imiquimod (TLR7) agonist and flagellin (TLR5 agonist) are widely known [80]. Flagellin, a protein from bacterial flagella, is recognized by Toll-like receptor 5 (TLR5) and activates innate and adaptive immunity [81]. In vivo experiments have shown that CBLB502, a preparation based on *Salmonella enterica* flagellin, can stimulate a stable antitumor response by directly activating TLR5-expressing immune cells, which, in turn, activate cytotoxic lymphocytes [82]. A combination of the administration of flagellin and CpG-containing oligodeoxynucleotides, which are also associated with pathogenic danger signals, completely suppressed tumor growth in mouse cancer models [83].

The intratumoral synthesis of danger signals due to gene therapy. For example, the synthesis of flagellin by genetically modified tumor-specific T-cells was shown to contribute to an increase in the antitumor effectiveness of immunity [84]. In a previous study [85], an adenovirus expressing fusion flagellin and chaperone Grp170 induced a potent anti-tumor response against B16 melanoma cells, mouse prostate cancer, and colon carcinoma. The genes encoding heat shock proteins HSP-70 and HSP-90 and the transcription factor HSF1 were also used as the DAMP signal encoded by oncolytic viruses [86]. It was shown that such viruses can induce a specific immune response in melanomas and colorectal and cancerous tumors in immunocompetent mice [87]. Preclinical trials of the HSP70-overexpressing oncolytic virus demonstrated the inhibition of primary and metastatic tumors due to enhanced oncolytic activity and an HSP-mediated immune response when administered intratumorally. Phase I clinical trials have demonstrated antitumor efficacy in regressing some distant metastases in certain patients and increasing the number of immune CD4+ and CD8+ T cells and NK cells [88].

Clinical evidence suggests that the induction of ICD or danger signals alone is not sufficient to overcome the immunosuppressive tumor microenvironment [89,90]. Cell death and the release of DAMPs may also trigger chronic inflammation and thereby promote the development or progression of tumors [91]. Thus, to increase the probability of therapeutic success, it is desirable to introduce another component into the system that also contributes to the maturation of dendritic cells, alongside the presentation of antigens and the recruitment of T-lymphocytes. GM–CSF can act as such a component. GM–CSF plays an important role in DC recruitment and maturation but also facilitates the homing of cytotoxic lymphocytes (CTLs) in the TME. Different preclinical studies showed that the intratumoral expression of GM–CSF yields an effective anti-tumor immune response [92]. Multiple vaccine platforms include GM–CSF in their formulations, and the goal of intratumoral injection is to increase the number of DCs in TME [93]. In the following section, the immune roles of GM–CSF, danger signals, and ICD in various tumor treatments are discussed (see the below section and Figure 2).

## 5. GM–CSF, Danger Signals, and ICD’s Role in the Immune Effects of Various Tumor Therapies

### 5.1. Chemotherapy

Chemotherapy can yield tumor immunity in two ways: by inducing ICD and by preventing the evasion of tumor immune surveillance [94]. Some chemotherapy drugs, particularly anthracyclines (e.g., idarubicin, epirubicin, doxorubicin, and mitoxantrone) mediate their antitumor effects at least in part by inducing ICD, thus releasing TAAs and DAMPs (e.g., PRR, IFNs, ATP, CCL2, CXCL1, and ANXA1) in TME (Figure 2A). The availability of TAAs and DAMPs in the microenvironment leads to the abundant recruitment of APCs and their precursors, the significant functional activation of APCs, the recruitment of T cells, and the effective phagocytic uptake of dying cancer cells and their fragments. Finally, APCs present TAA-derived epitopes to CD4+ and CD8+ T cells to provide TAA-targeted immunity [95]. Current trends in the preclinical and clinical development of ICD-eliciting chemotherapy as a partner for immunotherapy is described in [95]. Whether cell death from therapy is immunogenic (ICD) or not depends on the tumor’s biology and chemotherapeutic agent being used [6,94,96]. GM–CSF can be used to enhance recognition, thereby releasing tumor antigens from immune cells. A description of the effects of GM–CSF’s application in various chemotherapies can be found in [56]. Moreover, GM–CSF (Sargramostim) is approved in the USA for the prevention and treatment of chemotherapy-induced neutropenia and hematopoietic stem cell mobilization. A meta-analysis [97] based on the results of 15 clinical studies confirmed that GM–CSF can help a cancer patient, after chemotherapy treatment, quickly restore his or her white blood cells and absolute neutrophil counts. There are no data on the immunological status of the patients studied in this meta-analysis, but the overall results of the hematological indexes look promising.

### 5.2. Radiotherapy

Radiotherapy induces DNA damage and consequential cell death (e.g., ICD) and causes changes in the immune composition of the tumor, thereby leading to dendritic cell activation (Figure 2A). However, these changes can also increase the relative ratio of Treg to CD4+ cells, which slows the effective immune response [98]. A combination with immune therapy could potentially overcome the inhibitory pool of regulatory T cells through CTLA-4 inhibition; however, many challenges still remain [96,99]. Moreover, it was recently shown that local radiotherapy and GM–CSF in combination can generate an abscopal effect (a measurable response at the distant localization of a tumor or metastasis after local treatment) in patients with metastatic solid tumors [100,101,102].

### 5.3. Vaccines

Self and modified antigens of tumor cells were used as epitopes for immunization in the design of anti-tumor vaccines. However, these antigens were shown not to be efficient enough to generate a sufficient T cell antitumor response. The inclusion of foreign antigens, such as influenza hemagglutinin as DAMP, into cancer vaccines could overcome this obstacle [103]. These antigens mimic pathogenic particles, thereby providing danger signals and facilitating uptake of the vaccine by DCs, as well as their maturation and migration to the lymph nodes [61,74,104], where DCs present the antigen to the T cells (Figure 2B). Additionally, GM–CSF has been used as a vaccine adjuvant [52,105]. Many tumor vaccines use irradiated tumor cells engineered to produce GM–CSF (see, for example, [106]). However, the results of applying GM–CSF as a vaccine adjuvant were not always beneficial—sometimes, immune suppression was enhanced. Further investigations are needed to determine appropriate dosages and patient populations for efficient GM–CSF applications [52,107].

### 5.4. Oncolytic Viruses

Oncolytic viruses (OVs) are genetically engineered or naturally occurring viruses that selectively replicate and kill cancer cells without harming normal tissues (reviewed in [108,109,110,111]). Immunogenic apoptosis, necrosis, and autophagic cell death caused by OVs, release a repertoire of TAAs, danger signal molecules (DAMPs and OV-derived PAMPs), and inflammatory cytokines, which evoke anti-tumor immunity (Figure 2C). Ideally, a strong immune memory will then recognize and attack tumor cells in the case of a relapse or the development of metastatic tumors at any anatomical location in the patient. OV vectors have been constructed to express various TAAs in the tumor during infection, which might increase immune responses against these TAAs. Oncolytic viruses, moreover, provide a number of potential advantages over conventional cancer therapies. For example, they possess higher cancer specificity and better safety, and OV-mediated oncolysis releases key signals to DCs and other APCs, which initiate a potentially potent antitumor immune response [112]. However, OV-mediated immunotherapy suffers from a number of challenges, including the relative inefficiency of delivering OVs to the tumor, slow viral replication within the tumor mass, and highly immunosuppressive TME [64]. OVs can be genetically engineered to express death-pathway-modulating genes, thus causing ICD and enhancing immunogenicity [64,113,114,115]. The GM–CSF gene is often artificially inserted into OV genomes to disrupt immune tolerance and produce antitumor immunity [111]. Other immunomodulatory genes have also been inserted into OVs, including IL-4, IL-12, IL-18, IL-24, RANTES, CD40L, CD80, IFN-α, and IFN-β) [110,111,112,116], which remain in preclinical or clinical trials [117].

The first oncolytic virus approved by the FDA for the treatment of cutaneous and subcutaneous melanoma was T-VEC (in 2015). T-VEC (Talimogene laherparepvec; ImlygicTM) is a genetically modified herpes simplex virus (type 1) engineered to express GM–CSF and provides great information on the efficiency and safety of OV applications in clinic. The effect of T-VEC is attenuated by the deletion of herpes neurovirulence viral genes enhanced for immunogenicity by the deletion of the viral ICP47 gene. Immunogenicity is supported by expression of the GM–CSF gene. T-VEC yielded a significant improvement in the durable response rate, objective response rate, and progression-free survival in a randomized phase III clinical trial for patients with advanced melanoma [75,118,119,120]. Relative to GM–CSF, T-VEC significantly increased the durable response rate (16.3 vs. 2.1%, *p* < 0.001). However, the median overall survival was improved only slightly (23.3 vs. 18.9 months, *p* = 0.051). Phase 1b trials have combined T-VEC and immunotherapy with promising results. T-VEC therapy has yielded several important management considerations of T-VEC as a live virus, such as the need for appropriate storage and handling of the agent and problems related to virus biosafety control, administration, and the prevention of household contact transmission [118]. Currently, several viruses, including those that have been genetically engineered, are being studied in clinical trials (up to phase 3), but it remains too early to judge their clinical merits and shortcomings.

## 6. Opportunities for Combinations of Chemo-, Radio-, Suicide-, and Immunotherapy

Successes in cancer treatment with immunotherapy have been impressive. However, monotherapy entails significant limitations. Therefore, there is an increasing interest in using this strategy in combination with more traditional treatments, such as chemotherapy, radiation therapy, and molecularly targeted therapy. Ongoing trials of various therapeutic combinations (reviewed in [8,94,96]) have demonstrated improvements in therapeutic outcomes. The most important phenomena observed in these treatments were the synergistic effects of such combinations. For example, a combination of cancer vaccines with checkpoint inhibitors led to synergistic effects and higher response rates than monotherapy [8], and clear synergism was reported in the treatment of non-small-cell lung cancer with radiotherapy and a PD-1 antibody [121]. The synergistic effects of different immune therapies, such as dendritic cell vaccination and CTLA-4 blockade combined with irradiation, were described in [99].

Recently, we reported the antitumor efficacy of combined suicide gene therapy and radiotherapy in a model of CT26 murine colon adenocarcinoma [122]. In this study vector with FCU1 and GM–CSF genes in PPT, a delivery shell was used for intratumoral administration. The FCU1 gene encodes a fusion protein of yeast CD and yeast uracil phosphoribosyltransferase (UPRT), which catalyzes the conversion of a prodrug (5-FC) into a toxic drug (5-FU monophosphate)—a function frequently diminished in tumor cells, rendering them resistant to 5-FU treatment. In this study, tumors were irradiated with a single 5 Gy dose. A pronounced antitumor effect of the FCU1-GM–CSF-PPT/5-FC system combined with radiotherapy was observed under the background of a prolonged lifespan and the synergism of the applied methods. Earlier attempts to combine gene therapy and radiotherapy were described in [123].

All the pre-clinical results mentioned above appear to be promising. However, many challenges remain to be investigated.

## 7. Regulation of Expression and Optimization of the Delivery System in Support of Anticancer Gene Therapy

In addition to using various combinations of therapeutic genes and combinations with other types of anticancer therapy, the efficiency of gene therapy could be increased by fine-tuning the therapeutic gene expression and optimizing the gene-delivery system. The use of promoters with different strengths and specificities, in theory, could facilitate enhanced control over the expression of transgenes. In addition, the choice of the delivery system is very important since each system has a number of advantages and disadvantages.

### 7.1. Expression Regulatory Element

The promoter/enhancer regulatory element controls expression of the therapeutic gene. This element could drive tumor-restricted gene expression, thus improving the safety profile. This element could also provide a high enough level of expression to produce sufficiently large quantities of the enzyme to induce immunogenic cell death and sufficient levels of the danger signal and GM–CSF to ensure interactions with dendritic cells. Frequently, natural tissue- and tumor-specific promoters are used for this purpose. The list of such promoters is continuously expanding. However, these promoters have two important drawbacks. First, they are relatively weak compared to, for example, strong constitutive CMV or SV40 promoters. Second, most promoters are active only in a few cancer-cell types. A perfect universal cancer-specific promoter would theoretically work in many different tumors but not in normal cells. Moreover, this promoter would successfully work not only in the primary tumor but also in its metastases. We previously developed spliced promoters of human telomerase and human surviving genes (PhTERT and PhSurv, respectively), which appeared to be universal and strong cancer-specific promoters [124]. We showed that the efficiency levels of the tandem promoters PhTSurv269 and CMV in driving the therapeutic genes of HSVtk and mouse GM–CSF are comparable in vivo. Unlike the CMV promoter, these promoters are tumor-specific and active in a wide range of cancer cell lines; we believe that their use to control the expression of therapeutic genes in anticancer gene therapy drugs will offer low toxicity due to the activity of such promoters only in cancer cells. Moreover, the constitutive PCNA promoter is a potentially universal promoter for gene therapeutic constructs since it has shown a relatively high and stable level of activity in cell lines of various origins [125,126]. Attempts to construct such an ideal promoter remain ongoing. Increased expression can also be achieved by increasing the translation efficiency and stability of mRNA [127].

### 7.2. Non-Viral Gene-Delivery System

Non-viral vectors are advantageous due to their low immunogenicity, practically unlimited packaging capacity for genetic material, and simple and low-cost production, thereby making them potentially safer in clinical use and more suitable for large-scale production [128,129,130]. However, these vectors are characterized by low gene-transfer efficiency and transfection of cells compared to viral vectors. A more detailed discussion of the advantages and disadvantages of viral and non-viral delivery systems in gene therapy can be found in recent reviews [11,12,131,132]. For this reason, non-viral delivery systems are significantly underestimated at present. The latest data on GDEPT systems that are now in clinical trials can be found in the database “Gene therapy. Clinical trials worldwide”, provided by the *Journal of Gene Medicine* (https://a873679.fmphost.com/fmi/webd/GTCT; updated in 2021, accessed on 24 August 2021). According to the database, there are 8 Phase III, 85 Phase II, 61 Phase I/II, and 218 Phase I registered clinical trials for the treatment of various cancers with suicide genes in various vectors. Adenoviral, retroviral, and herpes simplex type 1 viral vectors were used in 80, 96, and 15 trials, respectively. Only a few trials used non-viral delivery systems: 17 trials used naked plasmid DNA, and 3 trials used lipofection. In our study [10], we used the treatment of animals with an HSVtk/GM–CSF combination in one plasmid vector and a PEG-PEI-TAT peptide (PPT) copolymer for delivery. The system was, in most cases, able to inhibit the growth of the tumor and metastases, as well as increase the animal’s lifespan. The data demonstrated effective antitumor effects when using a simple and low-toxic non-viral carrier containing a vector with both suicidal and immunomodulatory genes.

The fact that non-viral transgenic delivery systems are extremely promising is reinforced by the use of such systems in modern mRNA-based vaccines against SARS-Cov-2 coronavirus. Two RNA vaccines—one from the US pharmaceutical giant Pfizer and BioNTech in Mainz, Germany, and another from Moderna in Cambridge, Massachusetts—were granted emergency approval from regulators in several countries to fight COVID-19. At least six RNA-based COVID-19 vaccines have entered human testing, and several more are nearing the clinic [133]. The systems of all these vaccines use lipid nanoparticles (LNPs) as a platform [134]. LNPs encapsulating mRNA within a solid lipid structure are composed of four components: cationic or ionizable lipids for mRNA complexation, cholesterol to stabilize the nanoparticle, helper phospholipids to aid in formation and intracellular release, and PEGylated lipids to reduce non-specific interactions. This technique with appropriate modifications could be applied to create GDEPT systems.

Another way to improve the delivery efficiency of the GDEPT system is to use guiding elements on the non-viral envelope. The molecular targeting of tumors has been the subject of a long search, ranging from exploiting the enhanced permeability and retention (EPR) effect of tumors by injecting various nanoparticles [135] or bacteria [136] to producing RGD-fiber-modified adenoviruses specially modified for better tumor penetration [137]. Molecular targets in tumors, such as exposed collagen, were used to create Rexin-G [138], which uses a von Willebrand factor (vWF) to modify the virus envelope and thus target tumor and metastatic niches when injected intravenously. These improvements in viral delivery methods can also be applied to non-viral GDEPT systems.

In our studies, we used the PDGFRβ-binding peptide YG2 to determine the possibility of targeting tumor-associated fibroblasts, which are characterized by the expression of the PDGFRβ receptor [139]. Another example is the use of MC1SP peptide, a ligand specific for melanocortin receptor-1 (MC1R), in the composition of polyplexes based on PEI–PEG to target murine melanoma tumor tissue [140]. Another peptide used in our work, TAT peptide (GRKKKRRQRC), nonselectively increases the penetration of nanoparticles into cells. The TAT-attached PEI–PEG copolymer cannot be used for targeted delivery to a tumor when polyplexes are administered systemically, but it works well when administered intratumorally [10].

## 8. Conclusions: A Look at GDEPT’s Future Development

Figure 3 summarizes some possible strategies for further improving the GDEPT system vector.

Choosing promoter/enhancer regulatory elements to provide transcriptional targeting of transgene expression in certain tumor cells, such as directly targeting cancer cells or microenvironment cells—e.g., CAFs. The duration of expression of cassettes under the control of different promoters is also important, as is the level of their activity. Recently emerging revolutionary CRISPR technology for targeted genome editing can be used to create more efficient regulatory systems.

Using a prodrug converting gene. Above, we briefly discussed several suicide gene systems. The success of GDEPT is dependent on its functional components, the enzyme and the prodrug; enhancing both of these elements could improve the whole system. For example, using a system that is cytotoxic in quiescence, as well as actively dividing tumor cells (e.g., oxygen-insensitive nitroreductase) could activate nitroaromatic prodrugs of DNA-damaging products, including in non-dividing cells [141].

Using immunotherapy components, such as GM–CSF and danger signals. The advent of novel immunotherapy checkpoint agents has opened new avenues for the treatment of metastatic cancers and offers therapeutic modalities to prolong remission. Therefore, it seems reasonable that immune checkpoint therapy could be included among combined modalities. Numerous studies on combined treatments, some of which were considered in this review, are currently ongoing and are, in many cases, demonstrating highly promising results. However, despite important clinical benefits, checkpoint inhibition is associated with numerous side effects, including dermatologic, gastrointestinal, hepatic, endocrine, and other less common inflammatory events, which are likely caused by general immunological enhancement. Combinations with other agents such as cytokines and danger signals could establish anti-tumor immune responses. Placing genes coding for artificial danger signals alongside GM–CSF in the same vector would provide higher concentrations of both entities within a single limited intratumoral space during ICD and allow both to synergistically influence the maturation of dendritic cells. Capturing antigens and cellular DAMPs in combination with artificial danger signals and the participation of GM–CSF would induce the maturation of DCs and their migration to secondary lymphoid organs where they can activate naive T-cells [142].

Non-viral delivery systems have many advantages and limitations. The delivery system could be improved through the successful development and application of mRNA vaccines. The use of cell-targeting or delivery-enhancing guiding elements could also enhance the non-viral delivery system of GDEPT. Delivery also includes the administration route. It seems logical to assume that the side effects of GDEPT and immunotherapy could be reduced via intratumoral administration. However, upon local administration, the immunologic responses were observed at cancer sites distant from the treatment targets, including metastases. This idea was recently formulated in [143].

In light of the recent improvements in our understanding of the immune components in tumor progression and the proper choice of the above components, non-viral delivered GDEPT has been shown to be simple in production, safe, and inexpensive. These factors will enable GDEPT to take its rightful place alongside other anticancer drugs, most likely in combination with other pre-existing approaches/therapies.

## Figures and Tables

**Figure 1 ijms-22-09376-f001:**
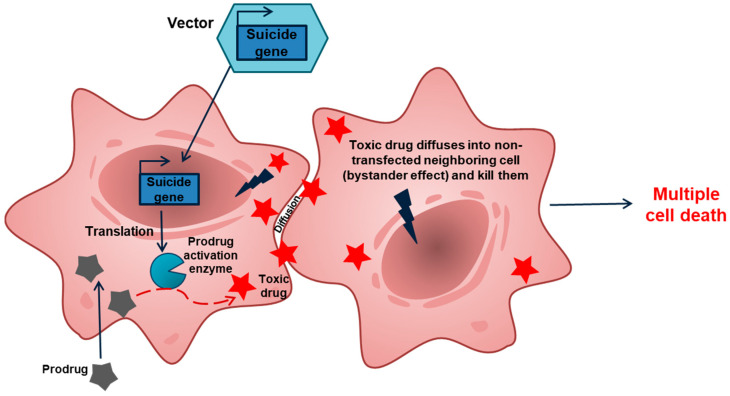
Overview of the GDEPT approach. Cancer cells are transfected with a construct carrying a suicide gene. Due to transfection, a fraction of cancer cells take up the gene, while other cells remain intact. Expression of the suicide gene within the cancer cells produces an enzyme that converts a prodrug into a toxic drug that leads to cell death. This toxic drug may diffuse into the surrounding cells (bystander effect) and kill them.

**Figure 2 ijms-22-09376-f002:**
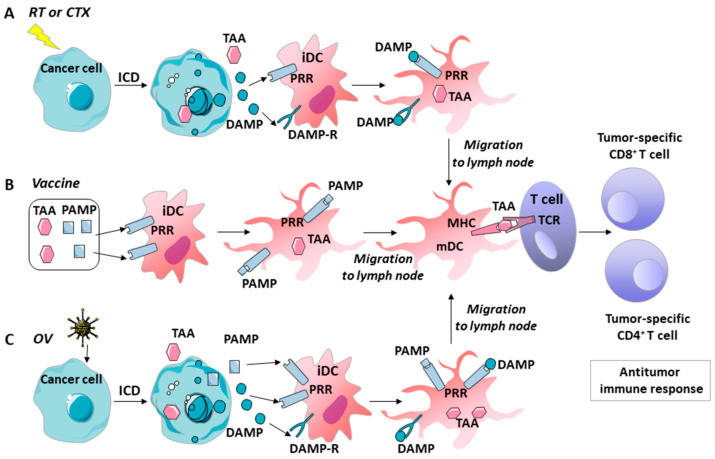
Activation of the antitumor immune response under various tumor therapies. (**A**) Radiotherapy or chemotherapy can be used to induce ICD in cancer cells. ICD results in the release of TAA and DAMPs in the tumor microenvironment. DAMPs are then recognized by PRRs and the DAMP receptors of tumor-resident immature DCs, which activates TAA uptake by DCs, DC maturation, and the presentation of TAA–MHC complexes. Then, DCs migrate to the lymph nodes to present TAA to immature effector T cells. Finally, the activated CD8+ and CD4+ effector T cells mediate the antitumor immune response. (**B**) A cancer vaccine was produced using a cocktail of TAA and foreign antigens. When vaccinated, foreign antigens act as PAMPs. These antigens provide danger signals to enable recognition of the vaccine by the PRRs of DCs, uptake of the vaccine TAA by DCs, and maturation of the DCs and their migration to the lymph nodes. Finally, activated CD8+ and CD4+ effector T cells mediate the antitumor immune response. (**C**) An oncolytic virus can induce ICD in cancer cells. ICD results in the release of TAA, as well as viral PAMPs and DAMPs in the tumor microenvironment. PAMPs and DAMPs are recognized by PRRs and the DAMP receptors of tumor-resident immature DCs, which activates the uptake of TAA by DCs, DC maturation, the presentation of TAA-MHC complexes, and migration to the lymph nodes. Finally, activated CD8+ and CD4+ effector T cells mediate the antitumor immune response. ICD, immunogenic cell death; TAA, tumor-associated antigens; DAMP, damage-associated molecular pattern; PAMP, pathogen-associated molecular pattern; PRRs, pattern-recognition receptors; DAMP-R, DAMP receptor; iDC, immature dendritic cell; mDC, mature dendritic cell; TCR, T cell receptor; RT, radiotherapy; CTX, chemotherapy; OV, oncolytic virus.

**Figure 3 ijms-22-09376-f003:**
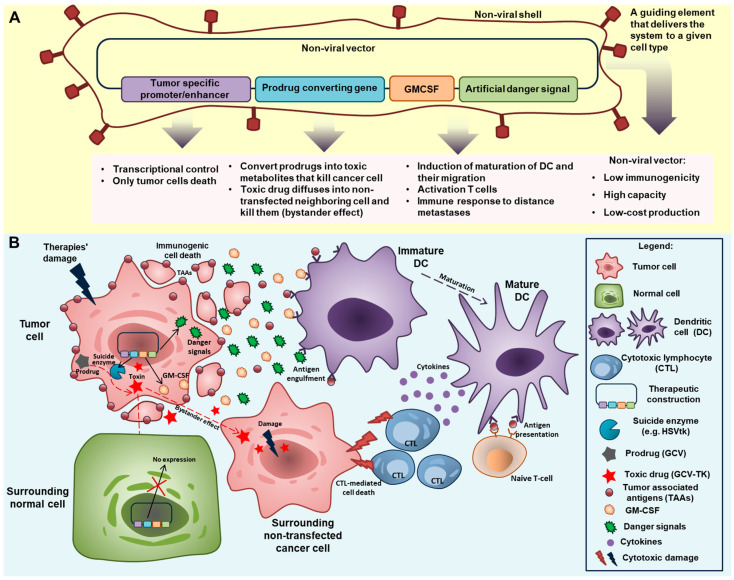
The “ideal” GDEPT system. (**A**) Components for improving the GDEPT system vector. A plasmid vector that encodes the suicide enzyme converting the prodrug to a toxin under the control of the tumor-specific promoter alongside GM–CSF plus an artificial danger signal as immunomodulators. The plasmid vector is coated by a non-viral shell with a guiding element that targets the system for a given cell type. (**B**) Mechanism of action of the “ideal” GDEPT system. A therapeutic construction administered intratumorally transfects the tumor cells because the guiding element of the non-viral shell and tumor-specific promoter/enhancer transgenes are restricted to expression in normal cells. In transfected tumor cells, the prodrug-converting gene (suicide enzyme gene) starts to be expressed and convert the prodrug (GCV or 5-FC) into toxic compounds (GCV-TK or 5-FU). The toxins cause apoptosis of the tumor cells and diffuse to the surrounding non-transfected tumor cells. Furthermore, this type of suicide gene therapy and its combination with chemo-/radiotherapies causes immunogenic cell death, thereby eliciting an immune response. Tumor-associated antigens are then released through immunogenic cell death and can activate dendritic cells and induce an adaptive antitumor response. GM–CSF and the danger signal co-expressed with the prodrug-converting gene enhance the activation of immunity. This combined impact leads to the induction of DC maturation and the migration of DCs to the tumor, the activation of T-cells, and the development of an immune response to distant metastases. All these factors lead to the effective destruction of tumor cells.

## Data Availability

Not applicable.

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
