# Peer review of "Step-by-Step Immune Activation for Suicide Gene Therapy Reinforcement"

_ijms, 2021, doi:10.3390/ijms22179376_

Round 1

Reviewer 1 Report

  1. In lines 80-84 the authors state that various enzyme-prodrug systems are in the stages of preclinical and clinical trials. Authors are invited to cite such studies, or at least cite a reference that addresses them.
  2. In lines 102-105 the authors state that there is significant progress in preclinical studies and clinical trials; however, they do not address what such progress consists of. For example, progress in clinical trials implies that this technology has reached phase 3, therefore those clinical trials that reached this phase should be discussed.
  3. At lines 372-373, the authors state: "However, these genes mostly remain in clinical or preclinical trials." Authors are invited to cite such studies, or at least cite a reference that addresses them.
  4. In lines 481-489, the authors establish a link on clinical trials that include GDEPT systems. Authors are invited to include a table which indicates the types of cancer addressed, the status of each of the clinical trials, codes of such clinical trials, and the sponsors of said studies, so that the reader can analyze said information and form your own criteria.

Author Response

We are grateful to the reviewer for the very thorough analysis of our manuscript. We agree with all the comments.

Point 1: In lines 80-84 the authors state that various enzyme-prodrug systems are in the stages of preclinical and clinical trials. Authors are invited to cite such studies, or at least cite a reference that addresses them.

Response 1: We created a table with a list of clinical trials of drugs containing the most commonly used suicide genes: herpes simplex virus thymidine kinase gene and the cytosine deaminase gene of Escherichia coli and yeast, based on the ClinicalTrials.gov database. We added a reference to this table to the sentence (lines 83-88).

The original text was:

Several of these systems are now in preclinical and clinical trials, including the most extensively studied systems of the herpes simplex virus thymidine kinase gene (HSVtk), with ganciclovir (GCV) as a prodrug, and the cytosine deaminase gene (CD) of Escherichia coli and yeast, which converts the prodrug antifungal agent 5-fluorocytosine (5-FC)—which features low toxicity—into the drug toxic 5-fluorouracil (5-FU).

The revised version is:

Several of these systems are now in preclinical and clinical trials, including the most extensively studied systems of the herpes simplex virus thymidine kinase (HSVtk), with ganciclovir (GCV) as a prodrug, and the cytosine deaminase (CD) of Escherichia coli and yeast, which converts the prodrug antifungal agent 5-fluorocytosine (5-FC)—which features low toxicity—into the antineoplastic antimetabolite 5-fluorouracil (5-FU) (See Supplementary Table 1).

Point 2: In lines 102-105 the authors state that there is significant progress in preclinical studies and clinical trials; however, they do not address what such progress consists of. For example, progress in clinical trials implies that this technology has reached phase 3, therefore those clinical trials that reached this phase should be discussed.

Response 2: We added to the text a description of the results of phase III clinical trials of the most well-known drugs based on the GDEPT systems HSVtk/GCV (Cerepro drug) and CD/5-FC (Toca 511 drug).

The original text was:

However, despite the great efforts, promising preliminary results, and long period time (over 25 years) since the conceptualization of this method [14, 26, 27] , as well as significant progress in preclinical studies and clinical trials, none of the various suicide gene therapy protocols have reached the clinic [14, 28].

The revised version is:

The first gene therapy trial using GDEPT system HSVtk/GCV was approved in 1991 [26]. Since then, a lot of clinical studies have been undertaken (Supplementary Table 1). At least three drugs based on HSVtk/GCV and CD/5-FC systems have reached the third phase of clinical trials: Cerepro® (developed by Ark Therapeutics Group), Toca-511 (developed by Tocagen), and ProstAtak® (developed by Candel Therapeutics, Inc).

Cerepro® (sitimagene ceradenovec) is an adenoviral vector encoding the herpes simplex thymidine kinase gene, for the treatment of patients with high-grade glioma, received the status of an orphan drug in 2002. In Phase I/II of clinical studies, sitimagene ceradenovec exhibited a significant increase in survival. Although the preliminary results of a Phase III clinical study demonstrated a significant positive effect of sitimagene ceradenovec treatment on time to reintervention or death when compared with standard care treatment (hazard ratio: 1.43; 95% CI: 1.06–1.93; p < 0.05), the European Committee for Medicinal Products for Human Use did not consider the data to provide sufficient evidence of clinical benefit [27, 28].

Toca 511 (vocimagene amiretrorepvec) is a nonlytic, retroviral replicating vector encoding the yeast cytosine deaminase. Toca 511 has shown highly promising results in early-stage clinical trials. In 2017, the European Medicines Agency granted the Toca 511 a priority review status, and the FDA designated it a “Breakthrough Therapy”. However, treatment with Toca 511 did not improve overall survival compared with the standard therapy in patients with recurrent high-grade glioma undergoing resection, missing the primary end-point of the phase III Toca 511 trial. Results showed that the median overall survival was 11.5 months and 12.2 months with Toca 511 and standard therapy, respectively (HR, 1.06; P = .6154). Moreover, all secondary endpoints did not show meaningful difference between the study arms [29]. Candel Therapeutics, Inc is currently conducting a Phase 3 trial for ProstAtak® (CAN-2409) in combination with the prodrug valacyclovir in patients with newly diagnosed localized prostate cancer who have an intermediate or high-risk for progression. It expects to complete enrollment in the 2021 with a final data readout in 2024.

Thus, despite the great efforts, promising preliminary results, and a long time (over 25 years) since the conceptualization of this method in 1986 [30,31], as well as significant progress in preclinical studies and clinical trials, none of the various suicide gene therapy protocols have reached the clinical setting until nowadays [14,32].

Point 3: At lines 372-373, the authors state: "However, these genes mostly remain in clinical or preclinical trials." Authors are invited to cite such studies, or at least cite a reference that addresses them.

Response 3: We added a link to a review by Taylor M. Pearl et al that looks at preclinical, early-stage clinical, and potential future efforts with cytokine-secreting oncolytic viruses (lines 412-414).

The original text was:

Other immunomodulatory genes have also been inserted into OVs, including IL-4, IL-12, IL-18, IL-24, RANTES, CD40L, CD80, IFN-α, and IFN-β) [108-110, 114]. However, these genes mostly remain in clinical or preclinical trials.

The revised version is:

Other immunomodulatory genes have also been inserted into OVs, including IL-4, IL-12, IL-18, IL-24, RANTES, CD40L, CD80, IFN-α, and IFN-β) [112-114, 118], which remain in preclinical or clinical trials [119].

Point 4: In lines 481-489, the authors establish a link on clinical trials that include GDEPT systems. Authors are invited to include a table which indicates the types of cancer addressed, the status of each of the clinical trials, codes of such clinical trials, and the sponsors of said studies, so that the reader can analyze said information and form your own criteria.

Response 4: Database “Gene therapy. Clinical trials worldwide” contains very limited data on any particular study. However, information from this database can be used to understand trends in cancer gene therapy. We created a table with a list of clinical trials of drugs containing the most commonly used suicide genes: herpes simplex virus thymidine kinase gene and the cytosine deaminase gene of Escherichia coli and yeast, based on the ClinicalTrials.gov results database. ClinicalTrials.gov has detailed information about each clinical study. We added a link to the table (lines 83-88).

We also corrected punctuation in the text and added Supplementary Materials Table S1: Clinical trials of HSVtk/prodrug and CD/5-FC systems.

All the changes are included in the “Ijms-1344095_corrections” file.

Reviewer 2 Report

This nice and well detailed review highlights the role and the impact of Gene-direct enzyme prodrug therapy (GDEPT) method for cancers treatment associated with classical known and well used treatments evolution over the last 25 years . The authors tackle the subject starting from the principle of using GDEPT in cancer treatment to induce a specific cancer cell suicide in the aim of targeting specifically the tumor with a good benefice-risk balance for the patients.

Minor comments suggestions:

  • In the 2nd title, the authors detail the evolution of GDEPT method over the past 25 years, to make the paragraph easy to get in, it would be better if a chronological schedule, time index, decade…etc. is integrated in the text, this will increase the dynamic of reading and allow to better appreciate the state of its development and uses.
  • The 3rd and 4th title/ parts could be merged in one, le first part of the 3rd Titled “A brief description of GM-CSF, the most commonly used immunomodulatory” could be more reduced and less detailed (useless) in the global context of the review.
  • In the last part of the review, the authors talk about the different methods used to regulate the expression and optimize the delivery system in support of anticancer gene therapy, what about/ what is the role of CRISPER technology now? To improve the insertion of the vector?

Author Response

We are grateful to the reviewer for the positive assessment of our work and valuable comments. We agree with the overwhelming majority of the comments, and only in one case (suggestion “The 3rd and 4th title/ parts could be merged in one”), we prefer to leave the situation unchanged. We hope the reviewer will not mind, since we are not talking about fundamental changes.

Point 1: In the 2nd title, the authors detail the evolution of GDEPT method over the past 25 years, to make the paragraph easy to get in, it would be better if a chronological schedule, time index, decade…etc. is integrated in the text, this will increase the dynamic of reading and allow to better appreciate the state of its development and uses.

Response 1: We added information about the date of the conceptualization of the GDEPT method and the date of the launch of the first clinical trial of the GDEPT system. We also added information about GDPET drugs that reached Phase 3 clinical trials and their dates.

The original text was:

However, despite the great efforts, promising preliminary results, and long period time (over 25 years) since the conceptualization of this method [14, 26, 27] , as well as significant progress in preclinical studies and clinical trials, none of the various suicide gene therapy protocols have reached the clinic [14, 23].

The revised version is:

The first gene therapy trial using GDEPT system HSVtk/GCV was approved in 1991 [26]. Since then, a lot of clinical studies have been undertaken (Supplementary Table 1). At least three drugs based on HSVtk/GCV and CD/5-FC systems have reached the third phase of clinical trials: Cerepro® (developed by Ark Therapeutics Group), Toca-511 (developed by Tocagen), and ProstAtak® (developed by Candel Therapeutics, Inc).

Cerepro® (sitimagene ceradenovec) is an adenoviral vector encoding the herpes simplex thymidine kinase gene, for the treatment of patients with high-grade glioma, received the status of an orphan drug in 2002. In Phase I/II of clinical studies, sitimagene ceradenovec exhibited a significant increase in survival. Although the preliminary results of a Phase III clinical study demonstrated a significant positive effect of sitimagene ceradenovec treatment on time to reintervention or death when compared with standard care treatment (hazard ratio: 1.43; 95% CI: 1.06–1.93; p < 0.05), the European Committee for Medicinal Products for Human Use did not consider the data to provide sufficient evidence of clinical benefit [27, 28].

Toca 511 (vocimagene amiretrorepvec) is a nonlytic, retroviral replicating vector encoding the yeast cytosine deaminase. Toca 511 has shown highly promising results in early-stage clinical trials. In 2017, the European Medicines Agency granted the Toca 511 a priority review status, and the FDA designated it a “Breakthrough Therapy”. However, treatment with Toca 511 did not improve overall survival compared with the standard therapy in patients with recurrent high-grade glioma undergoing resection, missing the primary end-point of the phase III Toca 511 trial. Results showed that the median overall survival was 11.5 months and 12.2 months with Toca 511 and standard therapy, respectively (HR, 1.06; P = .6154). Moreover, all secondary endpoints did not show meaningful difference between the study arms [29]. Candel Therapeutics, Inc is currently conducting a Phase 3 trial for ProstAtak® (CAN-2409) in combination with the prodrug valacyclovir in patients with newly diagnosed localized prostate cancer who have an intermediate or high-risk for progression. It expects to complete enrollment in the 2021 with a final data readout in 2024.

Thus, despite the great efforts, promising preliminary results, and a long time (over 25 years) since the conceptualization of this method in 1986 [30,31], as well as significant progress in preclinical studies and clinical trials, none of the various suicide gene therapy protocols have reached the clinical setting until nowadays [14,32].

Point 2: The 3rd and 4th title/ parts could be merged in one, le first part of the 3rd Titled “A brief description of GM-CSF, the most commonly used immunomodulatory” could be more reduced and less detailed (useless) in the global context of the review

Response 2: We are thankful to the reviewer for this recommendation. We shortened chapter three. However, we would rather not combine Chapters 3 and 4, because Chapter 3 deals with the biology of GM-CSF and its two-facedness in relation to cancer (GM-CSF can both promote and inhibit tumor progression); and Chapter 4 deals with danger signals and their possible influence on the behavior of GM-CSF in cancer.

The original text was:

GM–CSF is most often used as an immunomodulator in various antitumor therapies and can serve as a paradigmatic example of attempts to achieve nonspecific activation of the immune environment of the tumor. GM–CSF is a cytokine belonging to a group of colony stimulation factors (CSFs) that are produced at a low level by most tissues and cell types but whose production can be increased up to 1,000-fold by microorganisms, endo-toxins, or foreign cells [38]. The inducibility, lability, and short lifespans of CSFs allow them to function as highly efficient systems regulating hematopoietic cells homeostasis.[39] Specific membrane receptors for each CSF [40] appear at all stages of cell development in the granulocyte and monocyte–macrophage lineages.[39] GM–CSF (see [41,42]) is also a hematopoietic growth factor that controls differentiation of the myeloid lineage. GM–CSF is mainly produced by activated leukocytes in response to infection or injury and is capable of generating both granulocytes and macrophages from bone marrow progenitors, in addition to mediating their differentiation to other cell types participating in immune responses. GM–CSF can also modulate the function of mature hematopoietic cells [39] through transcription regulation in four known signaling path-ways—PI3K-Akt, ERK1/2, JAK2/STAT5, and NF-kB—that form an integrated network where the activity of one pathway interacts with the others.[43] In this way, GM–CSF can promote the growth, proliferation, and differentiation of numerous precursors and mature cell types. Knockout studies have demonstrated the largely redundant role of GM–CSF in maintaining granulocyte and macrophage populations in a physiologically steady state.[44]

The revised version is:

GM–CSF is most often used as an immunomodulator in various antitumor therapies and can serve as a paradigmatic example of attempts to achieve nonspecific activation of the immune environment of the tumor. GM–CSF is a cytokine belonging to a group of colony stimulation factors (CSFs) that are produced at a low level by most tissues and cell types but whose production can be increased up to 1,000-fold by microorganisms, endo-toxins, or foreign cells [42]. GM–CSF can promote the growth, proliferation, and differentiation of numerous precursors and mature cell types. Knockout studies have demonstrated the largely redundant role of GM–CSF in maintaining granulocyte and macrophage populations in a physiologically steady state [43].

Point 3: In the last part of the review, the authors talk about the different methods used to regulate the expression and optimize the delivery system in support of anticancer gene therapy, what about/ what is the role of CRISPER technology now? To improve the insertion of the vector?

Response 3: We thank the reviewer for this valuable advice. We added a phrase about CRISPR technology (lines 572-574) to the text of Manuscript.

The original text was:

Choosing promoter/enhancer regulatory elements to provide transcriptional targeting of transgene expression in certain tumor cells, such as directly targeting cancer cells or microenvironment cells—e.g., CAFs. The duration of expression of cassettes under the control of different promoters is also important, as is the level of their activity.

The revised version is:

Choosing promoter/enhancer regulatory elements to provide transcriptional targeting of transgene expression in certain tumor cells, such as directly targeting cancer cells or microenvironment cells—e.g., CAFs. The duration of expression of cassettes under the control of different promoters is also important, as is the level of their activity. Recently emerging revolutionary CRISPR technology for targeted genome editing can be used to create more efficient regulatory systems.

We also corrected punctuation in the text and added Supplementary Materials Table S1: Clinical trials of HSVtk/prodrug and CD/5-FC systems.

All the changes are included in the “Ijms-1344095_corrections” file.
